# RoMA: Scaling up Mamba-based Foundation Models for Remote Sensing

**Fengxiang Wang[1], Yulin Wang[2], Mingshuo Chen [3], Haiyan Zhao[2], Yangang Sun[2], Shuo Wang[2], Hongzhen Wang[2] , Di Wang[4,5]\*, Long Lan[1], Wenjing Yang [1]\*, Jing Zhang[4]\***

[1] College of Computer Science and Technology, National University of Defense Technology, China
[2] Tsinghua University, China [3] Beijing University of Posts and Telecommunications, China
[4] School of Computer Science, Wuhan University, China [5] Zhongguancun Academy, China

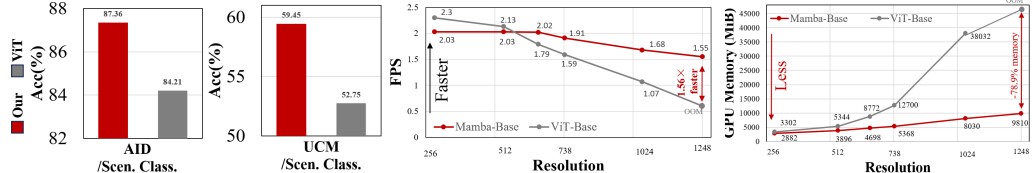

Figure 1: Comparison of ViT [1] (pretrained with MAE [2]) and our Mamba model (pretrained with RoMA) in scene classification, change detection, and semantic segmentation. Mamba outperforms ViT while being more computationally and memory efficient for high-resolution images. Notably, Mamba-B achieves $1.56\times$ faster inference and reduces GPU memory usage by 78.9% on $1248\times1248$ resolution images (6084 tokens per image) on a single NVIDIA 4090 GPU (batch size = 2).

## Abstract

Recent advances in self-supervised learning for Vision Transformers (ViTs) have fueled breakthroughs in remote sensing (RS) foundation models. However, the quadratic complexity of self-attention poses a significant barrier to scalability, particularly for large models and high-resolution images. While the linear-complexity Mamba architecture offers a promising alternative, existing RS applications of Mamba remain limited to supervised tasks on small, domain-specific datasets. To address these challenges, we propose RoMA, a framework that enables scalable self-supervised pretraining of Mamba-based RS foundation models using large-scale, diverse, unlabeled data. RoMA enhances scalability for high-resolution images through a tailored auto-regressive learning strategy, incorporating two key innovations: 1) a rotation-aware pretraining mechanism combining adaptive cropping with angular embeddings to handle sparsely distributed objects with arbitrary orientations, and 2) multi-scale token prediction objectives that address the extreme variations in object scales inherent to RS imagery. Systematic empirical studies validate that Mamba adheres to RS data and parameter scaling laws, with performance scaling reliably as model and data size increase. Furthermore, experiments across scene classification, changing detection, and semantic segmentation tasks demonstrate that RoMA-pretrained Mamba models consistently outperform ViT-based counterparts in both accuracy and computational efficiency. The source code and pretrained models were released at RoMA.

## 1 Introduction

Over the past decade, advancements in remote sensing (RS) technology and more efficient data acquisition have significantly enhanced applications in ecosystem monitoring [3], and natural disaster management [4]. These applications rely on crucial model capabilities such as scene classification [5],

---

*Corresponding authors

39th Conference on Neural Information Processing Systems (NeurIPS 2025).

object detection [6], change detection [7], and semantic segmentation [8]. However, training solely on limited task-specific data restricts the scale and generalizability of current RS deep learning models.

Recent breakthroughs in self-supervised learning (SSL) [2, 9] have led to the development of RS foundation models (RSFMs) [10, 11, 12, 13, 14, 15, 16] that offer robust feature representations and excel across various remote sensing tasks. However, many of these tasks involve high-resolution imagery—such as the 4,000×4,000 pixel images in the DOTA dataset for object detection. Most RSFMs rely on Vision Transformer (ViT)-based attention architectures, whose quadratic complexity limits their practicality on high-resolution data. To overcome this challenge, researchers are exploring pretraining RSFMs on architectures with linear complexity, with Mamba [17] emerging as a promising alternative.

The Mamba architecture is well-regarded in remote sensing for its efficient inference with high-resolution images in downstream tasks [18, 19]. However, current Mamba-based studies are limited to small-scale training datasets, restricting their exposure to diverse remote sensing data. This contrasts with trends in ViT-based RSFMs, which use self-supervised pretraining to harness extensive unlabeled data. Therefore, exploring self-supervised learning for Mamba to harness large-scale remote sensing data—and thereby compete with ViT—presents a promising, yet underexplored, direction.

Autoregressive pretraining [20, 21] offers a principled solution to Mamba's sequence continuity challenges by representing images as 1-D sequences and employing next-token prediction. Its causal token dependencies naturally align with Mamba's unidirectional linear-time scanning, preserving spatial coherence without the disruptions introduced by masking. While this approach has been successfully applied to natural images [20, 21, 22], RS images present unique challenges that remain largely unaddressed. We highlight three key challenges: (1) RS images often contain sparsely distributed foreground objects amid complex backgrounds. (2) Unlike objects in natural images, which typically maintain fixed orientations due to gravity, overhead RS images feature objects at varying orientations. (3) The wide range of object sizes in RS images complicates the extraction of effective representations. These challenges naturally lead to the question of whether Mamba-based RSFMs can scale efficiently with both increasing model size and larger data volumes—mirroring the performance improvements observed in self-supervised pretrained ViT architectures [23, 24].

To address these challenges, we propose Rotation-aware Multi-scale Autoregressive learning (RoMA), a framework that enables scalable self-supervised pretraining of Mamba-based RSFMs using large-scale, diverse, unlabeled data. Specifically, RoMA enhances scalability for high-resolution images through a tailored autoregressive learning strategy, incorporating two key innovations: (1) a rotation-aware pretraining mechanism combining adaptive cropping with angular embeddings to handle sparsely distributed objects with arbitrary orientations. By identifying key regions for rotation augmentation, it enhances rotation-invariant representation learning. Additionally, it embeds angle information during rotated cropping and requires the model to predict angular changes during autoregressive pretraining, further reinforcing rotation-invariant visual representations; and (2) multi-scale token prediction objectives that address the extreme variations in object scales inherent to RS imagery. By aggregating predicted token information across multiple spatial scales, this strategy helps Mamba capture more complete and structurally robust object representations during autoregressive pretraining.

Building on RoMA, we investigate its potential for pretraining Mamba-based RSFMs. Through systematic empirical studies, we confirm that Mamba aligns with RS data and parameter scaling laws, exhibiting reliable performance improvements as model and data size increase. Additionally, experiments across scene classification, changing detection, and semantic segmentation tasks show that RoMA-pretrained Mamba models consistently surpass its ViT-based counterparts in both accuracy and computational efficiency.

The contributions of this study are as follows:

(1) We introduce RoMA, the first self-supervised autoregressive pretraining framework for Mamba architectures in remote sensing, enabling efficient scaling to high-resolution RS imagery. RoMA validates that Mamba-based RSFMs follow scaling laws, achieving consistent performance gains with larger models and datasets.

(2) We propose a dynamic rotation-aware mechanism that integrates adaptive region cropping and angle-aware embeddings. By guiding the model to predict angular variations dur-

ing autoregressive learning, it effectively addresses rotational diversity and sparse target distributions, enhancing rotation-invariant feature learning.

(3) We design a multi-scale prediction objective that addresses the extreme variations in object scales, enabling the model to learn robust object representations for downstream tasks.

## 2 Related Work

**Remote Sensing Foundation Models.** While vast amounts of RS data exist, much of it remains unlabeled and thus inaccessible for supervised learning [25]. Recently, self-supervised learning frameworks have been employed to learn representations for tasks such as scene classification, object detection, and semantic segmentation, with methods falling into generation-based [2] and contrastive learning-based [26] categories. Notably, GASSL [27] and CACo [28] utilize spatio-temporal information, while SeCo [29] focuses on multiple Earth locations at different timestamps. Beyond representation learning, rotation-aware detection has also been widely studied in RS. ReDet [30] introduces a rotation-equivariant backbone, CSL [31] addresses angle boundary issues by turning regression into fine-grained classification, and S2A-Net [32] enhances detection accuracy by aligning features with rotated anchors. Most recent work in RS has primarily focused on Masked Image Modeling (MIM), categorized by general image knowledge [33], large parameter scales [34], spatio-temporal information [11], and multi-sensor data [35, 36, 37, 38], with multi-scale methods [13, 14, 15] improving performance. A recent study further explores plain ViT as a remote sensing foundation model by introducing a Rotated Varied-Size Attention (RVSA) mechanism to better handle arbitrarily oriented objects [39]. MA3E [40] incorporates angle factors into MIM training. In parallel, multimodal foundation models have emerged to bridge heterogeneous RS modalities. CROMA [41] combines contrastive radar–optical pretraining with masked reconstruction to learn rich multimodal RS representations, while AnySat [42] adopts a JEPA-based joint-embedding framework with scale-adaptive encoders to unify various resolutions, scales, and modalities. Despite these advances, most methods focus on ViT-based RSFMs and MIM pretraining, while the Mamba-based autoregressive models remain unexplored.

**Vision Mamba in Remote Sensing.** Recently, the Mamba architecture has excelled in NLP and has been adapted to the vision domain to address visual problems. Vision Mamba (Vim) [17] uses Vim blocks, consisting entirely of Mamba layers, with forward and backward scanning to model bidirectional representations. Vmamba [43] incorporates Visual State Space (VSS) blocks, combining Mamba and 2D convolution layers, supported by a Swin Transformer [44]-like pyramid architecture. The vision Mamba architecture has also expanded into remote sensing, producing various Mamba-based projects, categorized into four types: classification, detection, segmentation, and others. For classification, SSMamba [45] and SpectralMamba [18] handle hyperspectral data, while RSMamba [46] focuses on visible light. Detection methods like ChangeMamba [19] and RSCaMa [47] focus on change detection. Segmentation methods include Samba [48] and RS-Mamba [49], which use Mamba alone. Despite the growing research in RS using Mamba, current work is limited to supervised training on small-scale datasets, not fully exploiting the vast RS data.

**Self-Supervised Learning in Vision.** Inspired by the success of self-supervised learning in NLP, visual self-supervised learning methods are thriving in three main categories: contrastive learning [26, 50], autoregressive learning [20, 51, 52], and Masked Image Modeling (MIM) [9, 2]. Current research on MIM focuses on regression targets and masking strategies. Various targets include discrete tokens [53], HOG features [54], deep features [55], and frequencies [56], have already been explored. However, MIM methods often face training issues while pretraining the Mamba architeture [22]. Recently, autoregressive pretraining in the visual domain has been explored. Most works, like VAR [21], have explored the application of autoregression in image generation. We are more focused on the work of autoregression in self-supervised pretraining. iGPT [20] first highlighted the potential of autoregressive pretraining as a general self-supervised visual representation strategy. SAIM [51], RandSAC [57] and AIM [52] explored further. These works mainly focus on pretraining the ViT series and have not explored pretraining the Mamba series. ARM [22] firstly explored the compatibility of autoregressive pretraining with Mamba on the ImageNet [58] dataset, but it has not considered the specific issues of the RS field, like the rotation-invariant representations and various sizes information in RS images Notably, while ARM has explored models of different sizes on natural images, it has not evaluated Mamba's autoregressive pretraining performance across varying data

scales. In the RS field, we are the first to establish the relationships between Mamba's pretraining performance with data volume, and model size .

# 3 Method

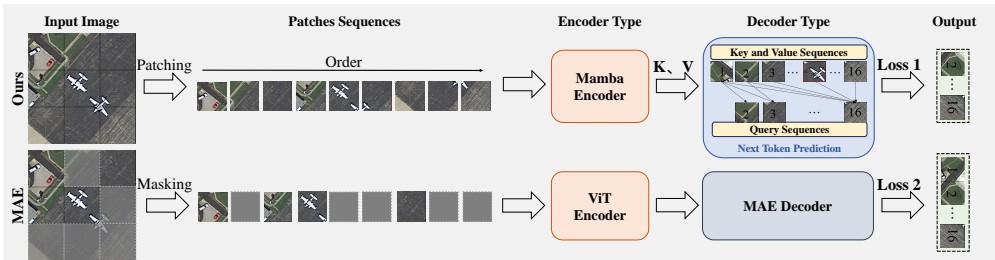

Figure 2: **Comparison between our autoregressive pretraining strategy and the standard MAE method.** (1) RoMA encodes all patches using a Mamba encoder, whereas MAE encodes only a randomly sampled subset. (2) RoMA predicts the next token in a sequence to capture continuity, while MAE only reconstructs masked patches.

## 3.1 Preliminaries

**Autoregressive Model.** Considering a sequence of discrete tokens $x = (x_1, x_2, \ldots, x_N)$, where $x_n \in [S]$ is an integer from a vocabulary of size $S$. The next-token autoregressive model posits that the probability of observing the current token $x_n$ depends only on its preceding tokens $(x_1, x_2, \ldots, x_{n-1})$. This unidirectional token dependency assumption enables the factorization of the sequence $x$'s likelihood as follows:

$$p(x_1, x_2, \ldots, x_N) = \prod_{n=1}^{N} p(x_n \mid x_1, x_2, \ldots, x_{n-1}). \tag{1}$$

Training an autoregressive model $p_\phi$ involves optimizing $p_\phi(x_n \mid x_1, x_2, \ldots, x_{n-1})$ over a dataset. This process, known as the next-token prediction, allows the trained $p_\phi$ to generate new sequences.

**MAE-based Pretraining of Mamba.** Previous work [10, 16, 11, 15, 13] primarily used MAE-based methods for pretraining Remote-Sensing Fundamental Models (RSFMs), where ViT is served as their visual backbones in often.

## 3.2 RoMA: Rotation-aware Multi-scale Autoregressive learning

We propose the RoMA autoregressive pretraining framework for the Mamba architecture in RS field. Specially, as shown in the Figure 3, we extend the iGPT [20] series with a $KV$ cache-based prediction method. The Mamba-Encoder processes the entire image to compute the $Key$ and $Value$ for all tokens. Then we calculate the learnable query vector from the $Key$ and $Value$, and compute the loss between the $Query$ and the target ground truth. Building on the autoregressive pretraining structure for natural images, RoMA introduces two key contributions: an adaptive rotation encoding strategy and a multi-scale prediction strategy.

### 3.2.1 Auto-regressive Pre-training of Mamba on RS Imagery

**Disadvantages of MAE-based Pre-training of Mamba.** First, we compare properties of general visual pre-training tasks and RS tasks, together with reflection on *why MAE-based pre-training are suboptimal choice towards RS imagery data:*

- **Explosion of visual tokens on high-resolution RS data.** As informed by Section 1, high-solution RS imagery exhibits numerous visual tokens. In contrast, the quadratic complexity of ViT-based MAT pre-training protocols is computational infeasible towards increasing visual tokens in high-resolution RS tasks (see detailed comparisons on speed, GPU usage and accuracy in Figure. 1).

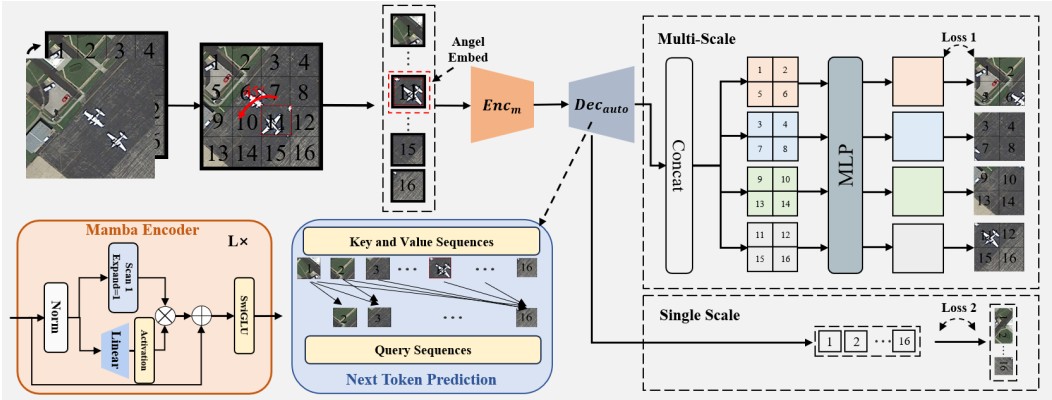

Figure 3: **Overview of the RoMA Pretraining Pipeline.** The input image is first divided into patches, and high-value patches are selected for random rotation using the Adaptive Rotation Encoding Strategy. These patches are then tokenized and processed by the Mamba encoder. The encoded features undergo autoregressive next-token prediction, followed by a multi-scale strategy that computes loss at different scales for gradient updates.

- **Disruption of MAE on Visual Tokens in Mamba's Architecture.** To be specific, MAE learns semantic representations by a first-masking-then-reconstructing pipeline. However, we observe that the mask operation in MAE disrupts between adjacent tokens, i.e., breaking sequential orders among tokens. As illustrated in Figure 2, the mask operation conflicts with the linear scanning operation embodied in Mamba, which aggregates temporally related tokens but not randomly related tokens.

**Advantage of Autoregressive Pre-training of Mamba.** Autoregressive pre-training aligns naturally with the sequential nature of Mamba's architecture, which processes input tokens in a temporally ordered manner. Specifically, autoregressive modeling constructs image patches sequentially and predicts the next token based on previous context, mirroring Mamba's token-by-token scanning mechanism. This architectural alignment facilitates more coherent temporal dependencies and better token transition modeling, allowing Mamba to learn more structured and semantically meaningful representations. Therefore, the autoregressive training paradigm not only complements Mamba's design but also enhances its ability to model spatial continuity and visual context in remote sensing imagery.

### 3.2.2 Adaptive Rotation Encoding Strategy

**Rotation-Invariant Pre-training is critical for RS Data.** As shown in Figure 4, RS images contain redundant airport runway pixels, and varying airplane angles lead to different postures and shapes. RS images often contain redundant airport runway pixels, while airplanes appear at various orientations, leading to different postures and shapes. Such directional diversity has been extensively addressed in supervised change detection through rotation-equivariant or rotation-invariant designs [30, 31, 32]. However,Autoregressive pretraining for natural images does not consider the uneven, sparse information distribution and rotational invariance in RS images. For instance, as shown in Figure 4, RS images contain redundant airport runway pixels, and varying airplane angles lead to different postures and shapes. These unique characteristics prompt us to rethink how the encoder can learn high knowledge density features with rotational invariance. In RoMA, we outline an adaptive rotation encoding strategy to enhance autoregressive pretraining for remote sensing. RoMA omits explicit angle prediction. Instead, angle embeddings introduce directional priors that help the model learn rotation-invariant representations during pre-training, without supervision from angle labels.

1. Split the input image.
2. Associate each patch $x^p$ with a score.
3. Selecting the patch (16×16) with the highest score.
4. Compute all 96×96 candidate boxes containing the patch and select the one with the highest

value.
5. Compare the 96×96 patch to the image-wide patch mean. If it exceeds the mean, select it; otherwise, proceed to 64×64 patches until one surpasses the mean.

We then detail each step outlined above for rotation-invariant encoding strategy: (1) **Step 1 of ARES:** We split the input image. $x \in \mathbb{R}^{H \times W \times C}$ into $N = (H \times W)/p^2$ non-overlapping patches $x^p \in \mathbb{R}^{N \times (p^2 C)}$, where $p$ is the patch size, $(H, W)$ is the size of the input image, and $C$ is the number of channels; (2) **Step 2 of ARES:** We associate each patch $x^p$ with a score, computed via a efficient feature descriptor $F$, e.g., LBP [59], and then select the token with the highest values; (3) **Step 3 of ARES:** Let $token_{top}$ denote the selected token. Then, centered on the patch represented by $token_{top}$, we expand its size to obtain a larger, more suitable region. (4) **Step 4 of ARES:** After extracting a square token region with an edge length of $L = \{96, 64, 32\}$, we compute its average feature value and compare it with the average feature value of each patch in the original image; (5) **Step 5 of ARES:** If $token_L$ has a higher average feature value, it is identified as a high-value region and proceeds to the rotation step. Otherwise, a smaller $L$ is selected, and step 4 is reapplied. This process repeats up to three times until a region with a higher average feature value than the original image is reached.

**Migrating Information Loss.** The selected patch is cropped to generate diverse rotated remote sensing scenes, while potential information loss on the edge pixels might occur. To mitigate this, we follow MA3E [40] and apply center cropping to retain the inscribed square (marked in yellow) within the largest inscribed circle. This region, oriented in any direction, replaces the original scene and introduces explicit angular variations. In addition to positional embeddings, we also incorporates learnable angle embeddings shared across patches within the rotated crop, i.e., served as implicit cues, aiding the model perceive angular changes while distinguishing them from the background.

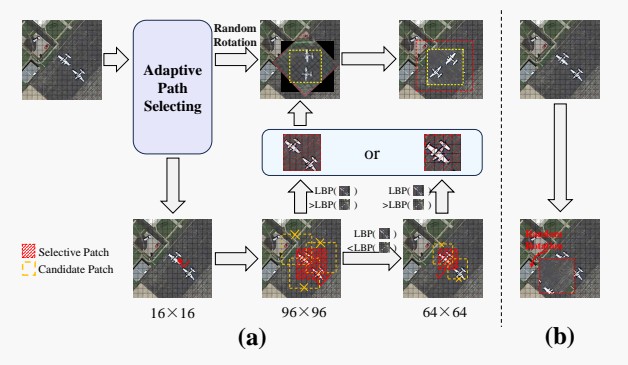

Figure 4: **Illustration of the Adaptive Rotation Encoding Strategy.** (a) Pipeline of the Adaptive Rotation Encoding Strategy. LBP refers to Local Binary Pattern. (b) Random patch selection for rotation without adaptive selection. The random approach in (b) disrupts object information in the RS image.

Finally, the Adaptive Rotation Encoding Strategy processes the image before feeding it into the Mamba-based encoder for representation learning. RoMA follows the standard Mamba architecture [22] without modifications, focusing purely on pretraining Mamba for RS field. While architectural improvements could enhance performance and efficiency, RoMA prioritizes pretraining strategies, leaving further Mamba optimizations to future research.

### 3.2.3 Multi-scale Prediction Strategy

Images are continuous 2D signals. To apply autoregressive pretraining via next-token prediction, two steps are required: (1) Convert images into discrete tokens. (2) Define them as a one-dimensional sequence for unidirectional modeling. Methods like iGPT [20] and VAR [21] tackle these challenges by slicing images into segments and arranging them into a feature sequence in a specified one-dimensional order.

However, as discussed in Section 1, directly applying this method to RS images fails to consider key factors. Unlike natural images, which focus on visual semantic understanding, RS images focus on surface measurement information [60]. Arbitrarily disrupting spatial relationships in RS images leads to fundamentally different interpretations. For example, token $x^{(i,j)}$ and its neighbors $x^{(i\pm1,j)}$, $x^{(i,j\pm1)}$ are closely related due to planar surface measurements. As seen in Figure 2, the token $x^{(i,j)}$

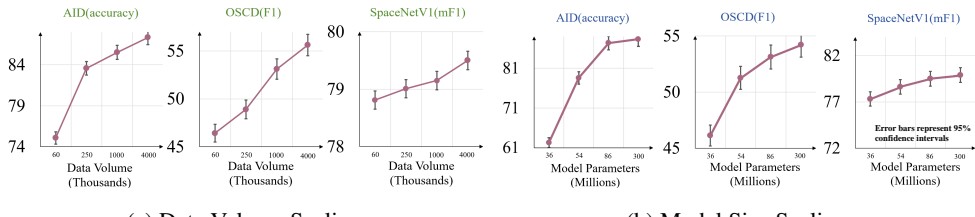

(a) Data Volume Scaling        (b) Model Size Scaling

Figure 5: **Scaling with Data Volume and Model Size.** Each experiment was conducted three times, and the average was reported as the final result. (a) We showcase the Mamba-Base model's performance on three downstream tasks after RoMA pretraining with different data scales. (b) We compare the performance of various Mamba model sizes on three downstream tasks, all pretrained with 4 million data using RoMA. Details on pretraining and downstream task configurations are provided in Section 4.

representing the part of plane is closely related to vertical neighboring tokens and some distant vertical tokens. The unidirectional flattening of autoregressive methods compromises surface information representation in RS images. Therefore, we introduce a multi-scale prediction strategy to mitigate the effects of unidirectional flattening.

The Mamba-based encoder generates $key$ and $value$ feature representations for each token. During decoding, we apply cross-attention with token-level causal masking to sequentially predict tokens, ensuring each token relies only on previously observed ones. The decoder performs autoregressive prediction at the token level. After the decoder generates the reconstructed outputs, the autoregressive module aligns them with the corresponding regions of the original image for supervision. The reconstruction quality is optimized using the mean squared error (MSE) loss between the predicted and original pixel values.

Building on MSE loss function, we concatenate token representations from each decoder block at a higher scale following a predefined raster order. A fully connected multi-layer perceptron (MLP) then reconstructs the pixel values of the next block. Notably, $x_k \in \mathcal{R}^{16 \times 16}$, while at a higher scale, spatial aggregation results in $y_n \in \mathcal{R}^{s \times s}$. $s$ is the pixel size value of larger scale. The formula is as follows:

$$\ell(\theta) = \frac{1}{K-1} \sum_{k=2}^{K} \|\hat{x}_k(\theta; x_{<k}) - x_k\|_2^2 + \frac{\lambda}{N-1} \sum_{n=2}^{N} \|\hat{y}_n(\theta; y_{<n}) - y_n\|_2^2 \qquad (2)$$

where $\theta$ represents the network parameters, $N$ is the number of cluster blocks in an image, $y_n$ is the ground truth pixel value of the $n$-th cluster block, and $\hat{y}_n(\theta; y_{<n})$ denotes the reconstructed value based on $\theta$ and preceding tokens ($y_{<n}$), $K$ is the number of all tokens in an image, $x_k$ denotes the ground truth pixel value of the $k$-th token, and $\hat{x}_k(\theta; x_{<k})$ is the reconstructed value based on the network parameters ($\theta$) and preceding tokens ($x_{<k}$) in the sequence. The parameter $\lambda$ regulates the contribution of the MSE loss from higher-scale cluster blocks to the overall loss.

### 3.3 Scaling Mamba-based RSFMs

To investigate the scaling potential of the self-supervised pretrained Mamba architecture for developing powerful RSFMs. With RoMA, we analyze the relationships between Mamba's performance with model parameters, and data scale. The scalability of ViT architectures pretrained with MAE has been well established, demonstrating performance gains with increasing data volume and model size [23, 24]. However, no prior work has systematically examined whether the Mamba architecture follows a similar trend in RS field. For the first time, we explore Mamba's scaling behavior in RS domain using the RoMA pretraining method.

Table 1: **The configuration of different architecture variants.**

| Model | Block | Width | Depth | Param.(M) |
|---|---|---|---|---|
| ViT-T | Attention+MLP | 192 | 12 | 5.7 |
| Mamba-T | Mamba+MLP | 192 | 12 | 5.3 |
| ViT-S | Attention+MLP | 384 | 12 | 22 |
| Mamba-S | Mamba+MLP | 384 | 12 | 21 |
| ViT-B | Attention+MLP | 768 | 12 | 86 |
| Mamba-B | Mamba+MLP | 768 | 12 | 85 |
| ViT-L | Attention+MLP | 1024 | 24 | 307 |
| Mamba-L | Mamba+MLP | 1024 | 24 | 297 |

Table 2: **Results for scene classification, change detection, and semantic segmentation.** "TR" represents the ratio of training data. ⋆ indicates results from MA3E [40] and MTP [61]. † denotes our reproduction with their official code.

| Methods | Publication | Backbone | Params | Scene Classification | | Change Detection | Semantic Segmentation |
| --- | --- | --- | --- | --- | --- | --- | --- |
| | | | | AID [62] | UCM [63] | OSCD [64] | SpaceNetv1 [65] |
| | | | | OA(TR=50%) | OA(TR=50%) | F1 | mF1 |
| *Natural Image pretraining* | | | | | | | |
| MoCo v3 ⋆ [50] | ICCV'21 | ViT-B | 86M | 78.72 | 38.34 | - | - |
| DINO ⋆ [26] | ICCV'21 | ViT-B | 86M | 78.51 | 40.04 | - | - |
| MAE ⋆ [2] | CVPR'22 | ViT-B | 86M | 84.21 | 52.75 | - | - |
| SimMIM ⋆ [9] | CVPR'22 | ViT-B | 86M | 83.19 | 51.48 | - | - |
| LoMaR ⋆ [66] | Arxiv'22 | ViT-B | 86M | 82.26 | 51.89 | - | - |
| MixMAE ⋆ [67] | CVPR'23 | Swin-B/W14 | 88M | 81.53 | 50.63 | - | - |
| ARM †[22] | ICLR'25 | Mamba-B | 85M | 81.14 | 50.41 | 47.28 | 77.89 |
| *RS Image pretraining* | | | | | | | |
| SeCo ⋆ [29] | ICCV'21 | ResNet-50 | 25.6M | 78.26 | 47.45 | 47.67 | 77.09 |
| CACo ⋆ [28] | CVPR'23 | ResNet-50 | 25.6M | 77.81 | 40.53 | 52.11 | 77.94 |
| SatMAE ⋆ [11] | NIPS'22 | ViT-L | 307M | 55.10 | 34.28 | 52.76 | 78.07 |
| ScaleMAE ⋆ [13] | ICCV'23 | ViT-L | 307M | 48.46 | 28.19 | - | - |
| GFM ⋆ [33] | ICCV'23 | Swin-B | 88M- | 80.58 | 49.73 | - | - |
| RVSA ⋆ [10] | TGRS'23 | ViT-B+RVSA | 86M | 84.06 | 50.86 | 50.28 | **79.56** |
| SatMAE++ † [15] | CVPR'24 | ViT-L | 307M | 85.98 | 55.72 | 53.10 | 79.21 |
| MA3E ⋆ [40] | ECCV'24 | ViT-B | 86M | 85.86 | 55.69 | - | - |
| RoMA | - | Mamba-B | 85M | **87.36** | **59.45** | **55.63** | 79.50 |

**Scaling with Data Volume:** Mamba shows a clear performance boost on downstream tasks as the pretraining data volume grows. We pretrain the Mamba-Base model with RoMA across various data scales and evaluate its performance in the downstream tasks. As illustrated in Figure 5a, larger datasets lead to significant improvements. Mamba-based RSFMs exhibit no significant performance bottlenecks across a broad pretraining data scale from 62.5K to 4M, achieving data learning capabilities on par with ViT-based RSFMs. We look forward to future advancements of data volume in remote sensing, where larger datasets can further enhance Mamba-based RSFMs through our RoMA pretraining framework.

**Scaling with Model Size:** Mamba's performance also improves with increasing model size. We conduct extensive pretraining on four model variants—Tiny, Small, Base, and Large—following the configurations in our code. As shown in Figure 5b, larger models consistently achieve superior results on downstream tasks. Although Mamba-Large surpasses Mamba-Base in AID dataset, its performance gain remains limited, likely due to insufficient pretraining. With only 300 epochs on 4 million samples, the training may not be adequate for a 297M-parameter model. Due to experimental constraints, we did not extend pretraining to 800 epochs as in MAE. The OSCD and SpaceNet experiments are ongoing, with updates to follow. However, these results do not alter our key findings: Mamba-based RSFMs pretrained with RoMA demonstrate performance gains as model parameters scale. While this growth remains inconclusive in more large-scale experiments, we anticipate future research will further explore Mamba's scaling potential.

## 4   Experiments

We pretrain Mamba extensively using RoMA and assess its effectiveness across diverse downstream tasks. Finally, we conduct thorough ablation studies on RoMA's design choices.

**Pretraining Setup.** Our pretraining experiment setup largely follows ARM [22]. We train both the Mamba-B on the OpticalRS-4M [16]. We adjust the input image to a size of $196 \times 196$, with a patch size of 16, using the AdamW optimizer and a cosine learning rate scheduler. The initial learning rate is set to 1.5e-4, and batch size is set to 256, with a epoch of 400.

**Downstream Tasks.** We further evaluated RoMA across three key downstream tasks: scene classification, changing detection, and semantic segmentation. In addition to benchmarking against ViT-based RSFMs, we compared RoMA with other pretraining methods for natural images. These encompass methods leveraging contrastive learning and generative learning and autogressive pretraining approaches, ARM [22]. The Mamba-B architectures strictly adhere to the simplest Mamba design from ARM, without any modifications, allowing us to exhaustively test the advantages of the RoMA pretraining framework.

Table 3: **Ablation study on the design choices of RoMA with Mamba-B backbone.** We report the top-1 accuracy (%). The default settings of RoMA are highlighted in grey.

(a) **Main ablation.** Adaptive Rotation Encoding Strategy (ARE) and Multi-scale Prediction Strategy (MSP) significantly improve RoMA.

(b) **Feature Descriptor** in Adaptive Rotation Encoding Strategy. Local Binary Pattern (LBP) measurement outperforms the Wavelet Transform and Histogram of Oriented Gradients (HOG).

(c) **Selecting Patch Size** in Adaptive Rotation Encoding Strategy. Three layers is the most effective choice.

| ARE | MSP | AID | |
|-----|-----|-----|-----|
| | | OA (TR=20%) | OA (TR=50%) |
| | | 69.59 | 76.80 |
| ✓ | | 71.70 | 78.00 |
| ✓ | ✓ | 72.69 | 79.16 |

| Feature Descriptor | AID | |
|-----|-----|-----|
| | OA (TR=20%) | OA (TR=50%) |
| Wavelet | 71.42 | 77.00 |
| HOG | 71.94 | 78.32 |
| LBP | 71.70 | 78.00 |

| Patch Size | AID | |
|-----|-----|-----|
| | OA (TR=20%) | OA (TR=50%) |
| 96 | 70.48 | 76.82 |
| 96-32 | 71.23 | 77.12 |
| 96-64-32 | 71.70 | 78.00 |

(d) **Threshold** for patch selection in the Adaptive Rotation Encoding Strategy is based on the image's overall average computed ($Avg.$) from the Feature Descriptor.

(e) **Coefficient** $\lambda$. The variation of the coefficient $\lambda$ in Multi-scale Prediction Strategy. $\lambda$ balances autoregressive reconstruction and sparsity regularization.

(f) **Various Scales** choices in Multi-scale Prediction Strategy. Experiments were conducted using the standard 16×16 patch as a baseline, with additional combinations at multiples of 2–6.

| Threshold | AID | |
|-----|-----|-----|
| | OA (TR=20%) | OA (TR=50%) |
| $1.5 \times Avg.$ | 68.33 | 72.39 |
| $0.5 \times Avg.$ | 69.71 | 74.18 |
| $1.0 \times Avg.$ | 71.70 | 78.00 |

| Coefficient $\lambda$ | AID | |
|-----|-----|-----|
| | OA (TR=20%) | OA (TR=50%) |
| 0.01 | 71.81 | 78.23 |
| 1.0 | 70.92 | 77.49 |
| 0.1 | 72.69 | 79.16 |

| Multi-scale Prediction Strategy | AID | |
|-----|-----|-----|
| | OA (TR=20%) | OA (TR=50%) |
| 2×+4×+6× | 63.17 | 71.34 |
| 2×+4× | 68.06 | 74.74 |
| 4×+6× | 67.81 | 74.32 |
| 4× | 72.46 | 78.72 |
| 6× | 72.69 | 79.16 |

**Scene Classification.** By freezing the model's parameters and fine-tuning only the final fully connected (FC) layer, linear probing effectively demonstrates its feature extraction ability. Since full-parameter fine-tuning already achieves over 99% performance on classification datasets, like AID [62], we prefer linear probing rather than fine tuning. We use two scene classification datasets: AID [62] and UCM [63], with training details, including the train-test split ratio, following [10, 13]. Evaluation is based on overall accuracy (OA). The results in Table 2 show RoMA's competitive performance compared to other pretraining methods.

**Change Detection.** We used the OSCD [64] dataset consisting of RGB images for change detection. Following previous works [61], we kept the experimental setups consistent, using UNet [68] as the decoder. On the OSCD dataset, our method outperforms ARM [22] and other methods that overlook rotational invariance and information sparsity and varying object size issues in RS.

**Semantic Segmentation.** We further evaluate the pretrained model on semantic segmentation tasks, using common remote sensing datasets: SpaceNetv1 [65]. Our implementation follows [61], using UperNet [69] as the segmentation framework. However, in pixel-level tasks, Mamba-based RSFMs show less pronounced advantages compared to other downstream tasks. We attribute this to RoSA's autoregressive pretraining, which prioritizes multi-scale patch-based targets over pixel-level prediction.

**Ablation Study.** Due to resource constraints, we conducted the ablation experiments using the MillionAID [70] dataset and trained for 400 epochs. Table 3a presents the performance of RoMA's two main contributions, with ARM as the baseline. The experiment shows a significant gain in feature extraction capability. Tables 3b and 3c adding the Adaptive Rotation Encoding Strategy on the baseline. Table 3b comparing the effects of different Feature Descriptors. We believe the Feature Descriptor method can be further optimized without affecting the Adaptive Rotation Encoding Strategy's effectiveness. Table 3c evaluates patch sizes for rotation, showing that various sizes improve performance. Tables 3d, 3e, and 3f examine the Multi-scale Prediction Strategy. The parameter and threshold selections are shown in Tables 3d and 3e, while Table 3f presents the performance of aggregating spatial features from multiple scales. Our results indicate that adding excessive multi-scale information doesn't guarantee improved performance; instead, using only large-scale aggregated information along with the original 16×16 patch data yields better results.

Table 4: **Peak GPU memory usage (MB) across different input resolutions.**

| Resolution | 768 | 1024 | 1248 | 1520 | 2048 | 3072 | 4096 |
|-----|-----|-----|-----|-----|-----|-----|-----|
| RoMA-Base | 2693 | 4504 | 6526 | 9434 | 16934 | 37485 | 66357 |
| ViT-Base | 4229 | 11726 | 24531 | 52106 | OOM | OOM | OOM |

Table 5: **Inference speed (samples/sec) across different input resolutions.**

| Resolution | 768 | 1024 | 1248 | 1520 | 2048 | 3072 | 4096 |
|-----|-----|-----|-----|-----|-----|-----|-----|
| RoMA-Base | 24.98 | 15.91 | 11.43 | 7.86 | 4.37 | 2.00 | 1.15 |
| ViT-Base | 22.11 | 9.94 | 4.99 | 2.57 | OOM | OOM | OOM |

# 5 Further Analyses

**Scalability to Ultra-High-Resolution Inputs.** To further examine RoMA's scalability, we evaluated it on inputs ranging from $768 \times 768$ to $4096 \times 4096$. Both RoMA-Base and ViT-Base were tested for GPU memory usage and inference speed on a single NVIDIA A100 (batch size = 1). As shown in Table 4 and Table 5, RoMA scales stably up to $4096 \times 4096$, while ViT-Base fails beyond $2048 \times 2048$. These results further verify RoMA's computational efficiency and suitability for ultra-high-resolution remote sensing imagery.

**Ability to Learn Small Targets.** To further analyze RoMA's ability to capture local information, we evaluate its performance on small-object categories in the iSAID dataset [71]. We compare UperNet [69] with different backbones, following the RingMo [72] fine-tuning protocol. As shown in Table 6.Our method achieves the highest overall mIoU and also shows notable improvements on small-object classes, especially *Small Vehicle* (average width 15 pixels), where RoMA surpasses all other backbones. These results suggest that the proposed **adaptive region cropping strategy** effectively increases the visibility of small foreground objects during pretraining, allowing the model to learn more discriminative representations.

Table 6: **Fine-tuning performance on iSAID [71] (following RingMo [72] settings).**

| Method | Backbone | mIoU | Ship ($33^2$ px) | Small Vehicle ($15^2$ px) | Swimming Pool ($34^2$ px) | Plane ($53^2$ px) |
|---|---|---|---|---|---|---|
| UperNet | IMP-ResNet-50 | 61.9 | 65.9 | 48.8 | 44.5 | 83.8 |
| UperNet | SeCo-ResNet-50 | 57.2 | 63.9 | 44.8 | 9.3 | 83.3 |
| UperNet | RSP-ResNet-50 | 61.6 | 64.2 | 47.5 | 43.8 | 82.8 |
| UperNet | ViT-B+RVSA | 63.8 | 68.9 | 51.9 | 46.7 | 85.6 |
| UperNet | ViTAE-B+RVSA | 63.5 | 69.6 | 51.9 | 47.5 | 85.4 |
| UperNet | RingMo | 67.2 | 73.5 | 51.2 | 48.9 | 85.7 |
| **UperNet** | **RoMA-B (Ours)** | **67.4** | **73.8** | **53.7** | **51.8** | **86.0** |

**Effectiveness of Top-$k$ Region Selection.** To further validate the effectiveness of the top-$k$ token selection mechanism, we conducted an additional experiment on the SpaceNet V1 [65] building segmentation dataset. This experiment aims to examine whether the proposed adaptive region cropping can consistently capture target regions compared with random cropping under the same resolution settings. As shown in Table 7, our method achieves a foreground capture accuracy of 75.09%, significantly higher than the 38.56% of random cropping. These results demonstrate that the top-$k$ strategy effectively preserves informative regions, leading to more reliable semantic representations for autoregressive modeling.

Table 7: **Foreground object capture rate on SpaceNet V1 [65].**

| Cropping Strategy | Accuracy (%) |
|---|---|
| Random Cropping | 38.56 |
| **Top-$k$ Adaptive Region Selection (Ours)** | **75.09** |

**Feasibility of Token-space Reconstruction Loss.** To evaluate the flexibility of RoMA, we compared pixel-space and token-space reconstruction losses. Following BEiT [53] and CAE [73], the token-space loss was computed using a frozen CLIP [74] teacher, where the student predicted its latent features through a lightweight MIM head. As shown in Table 8, token-space loss consistently outperforms pixel-space loss on UCM and AID, indicating its stronger semantic representation capability and the adaptability of the RoMA framework to diverse pretraining objectives.

Table 8: **Comparison between pixel-space and token-space reconstruction loss.**

| Pretraining Loss Type | UCM-55 (%) | AID-55 (%) |
|---|---|---|
| RoMA-Base (Pixel Space) | 52.39 | 80.34 |
| **RoMA-Base (Token Space)** | **56.67** | **81.72** |

# 6 Conclusion

We introduce RoMA, the first self-supervised autoregressive framework to scale Mamba-based foundation models for RS. By leveraging large-scale, diverse, unlabeled data, RoMA enables scalable self-supervised pretraining of Mamba-based RS models. Extensive experiments across scene classification, changing detection, and semantic segmentation show that RoMA-pretrained Mamba models consistently outperform ViT-based counterparts. Additionally, these models achieve significant efficiency improvements, reducing GPU memory consumption by 78.9% and accelerating inference speed by $1.56\times$ at $1,248 \times 1,248$ resolution, while maintaining linear computational scaling. Our findings also provide new insights into Mamba's scaling laws, demonstrating consistent performance gains with increasing data volume and model size, highlighting its potential for large-scale Earth observation tasks. **Limitations.** The current RoMA framework is evaluated mainly on optical imagery, and future work will extend it to multi-source remote sensing data (e.g., SAR and hyperspectral) .

**Acknowledgements:** This work was partially supported by the National Natural Science Foundation of China (No. 62372459, No.62376282 and No. 624B2109).

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
