# OpenReview forum: "RoMA: Scaling up Mamba-based Foundation Models for Remote Sensing"
_NeurIPS.cc/2025/Conference — NeurIPS 2025 poster_

### Official Review · Reviewer_c7wu · 2025-06-20

**Clarity:** 2
**Significance:** 2
**Originality:** 2
**Rating:** 3
**Confidence:** 3

**Summary:**

The paper seems to present novel designs and self-supervised pretraining strategies for foundation models in remote sensing, including the use of Mamba to avoid the quadratic bottleneck of transformer-based architectures, a patch rotation augmentation mechanism and an unclear multiscale prediction mechanism. However, the rather poor quality of writing makes it difficult to understand the actual contributions of the paper.

**Questions:**

The paper might have some interesting contributions from a technical standpoint but the presentation is too confusing. The authors should be significantly more careful in explaining the technical details of the method, clearly highlighting how every step of the proposed design and methodology works.

**Ethical Concerns:**

["NO or VERY MINOR ethics concerns only"]

**Final Justification:**

The work might present some interesting results by adapting Mamba to remote sensing foundation models but with modest methodological novelty. Clarity of manuscript writing remains a significant concern even after rebuttal.

**Limitations:**

Despite what is reported in the checklist, limitations are not discussed with the conclusions of the paper.

**Paper Formatting Concerns:**

Formatting is ok.

**Quality:**

2

**Strengths And Weaknesses:**

Strengths:
- The authors seek to improve the efficiency of foundation models in remote sensing. This is important to handle the very large images typical of this field.
- Experimental results seem promising with good metrics across different tasks compared to existing methods and significant runtime and memory savings

Weaknesses:
- The paper is very difficult to read due to poor writing quality. This makes it difficult to properly understand what the contributions of the work are and their technical details. There are several mistakes in the use of English (examples: caption of figure 2, lines 154-156, ...). A few examples of poor writing quality are: repetitions in lines 75-80; paragraph "Advantage of Autoregressive Pre-training of Mamba" does not explain any advantages, only the fact that a causal autoregressive model such as Mamba matches the setting of a causal autoregressive prediction task used for training.
- Several sections are unclear., particularly section 3.2.3. How are patches tokenized? what's the ordering that is decided for next-token prediction? It is said that the Mamba-based encoder (which is never explained what it is) generates key and values for a causal cross-attention operation. This is quite confusing. Not only it is not clear how it works, but also, this operation would have the classic quadratic complexity in the number of tokens of the transformer, which the authors claim is a main limitation of existing RS foundation models (lines 5,33,157). The "multi-scale" part of the prediction is also unclear: how are the multiple scales generated and processed? Neither the text nor fig.3 are illuminating on this aspect.
- The novelty of using a rotation augmentation with angular embedding for the pretraining method seems fairly limited

---

> ### Author Rebuttal · Authors · 2025-07-31
>
> ### **Q1:**  Poor writing obscures the contributions and includes grammar issues, repetition, and unclear sections.
>
> **A1:** We sincerely appreciate the reviewer's feedback regarding writing clarity. We acknowledge that certain sentences may lack fluency or appear ambiguous. In response:
>
> - **For the caption of Figure 2**, we agree that the phrasing can be improved. The current version indeed contains a typo ("different" → "difference", "autogresive" → "autoregressive").
>
>     In the revised version, we will rephrase it as:
>
>     *Comparison between our autoregressive pretraining strategy and the standard MAE method. (1) RoMA encodes all patches using a Mamba encoder, whereas MAE encodes only a randomly sampled subset. (2) RoMA predicts the next token in a sequence to capture continuity, while MAE only reconstructs masked patches.*
>
> - **For lines 154-156**, we acknowledge the original phrasing is grammatically flawed and may seem vague. In the revised version, we will rephrase this section for better clarity:
>
>     ***Limitations of MAE-based Pretraining for Mamba.** We begin by analyzing why MAE-based pretraining may be suboptimal for Mamba-based RSFMs, given the unique characteristics of RS imagery and the architectural mismatch between MAE and Mamba.*
>
> - **For lines 75-80**, which enumerate the contributions, the repetition noted by the reviewer may stem from syntactic redundancy. We will streamline the phrasing to make each contribution more concise, while preserving the technical depth. For example, we will replace the first contribution with
>
>     *(1) We introduce RoMA, the first autoregressive pretraining framework for Mamba in remote sensing, which supports high-resolution imagery and demonstrates consistent performance gains with increasing model and data scales.*
>
>     While the second contribution will be replaced by:
>
>     *(2) We propose a rotation-aware mechanism that combines adaptive region cropping with angular embeddings to enhance rotation-invariant representation learning. This design effectively handles orientation variability and sparsely distributed targets in remote sensing scenes during autoregressive pretraining.*
>
> - **Regarding the "Advantage of Autoregressive Pre-training of Mamba" paragraph**, our intent was to highlight the architectural alignment between Mamba and autoregressive modeling, as both employ sequential processing. We have revised this section to more clearly explain how this alignment brings practical benefits to the pretraining process. The paragraph will be revised to:
>
>     *Autoregressive pre-training aligns naturally with the sequential nature of Mamba's architecture, which processes input tokens in a temporally ordered manner. Specifically, autoregressive modeling constructs image patches sequentially and predicts the next token based on previous context, mirroring Mamba's token-by-token scanning mechanism. This architectural alignment facilitates more coherent temporal dependencies and better token transition modeling, allowing Mamba to learn more structured and semantically meaningful representations. Therefore, the autoregressive training paradigm not only complements Mamba's design but also enhances its ability to model spatial continuity and visual context in remote sensing imagery.*
>
>
> ### **Q2:** Section 3.2.3 is unclear, especially regarding patch tokenization, token order, multi-scale prediction, and the role of the Mamba encoder.
>
> **A2:** We thank the reviewer for pointing out the ambiguity in Section 3.2.3. We appreciate the opportunity to clarify the technical design regarding patch tokenization, sequence ordering, the Mamba encoder, causal decoding, and the multi-scale prediction mechanism.
>
> **Notably, many of these technical components follow widely adopted practices in the community (e.g., ViT-style patching, raster ordering, and standard Mamba configuration). To avoid redundancy, we did not elaborate on them in detail in the manuscript. However, we acknowledge the need for clearer exposition and will revise the manuscript accordingly.**
>
> Below, we provide a detailed response:
>
> - **Patch Tokenization**: Each input image is divided into non-overlapping patches (e.g., 16×16 pixels), which are then linearly projected into visual tokens via a trainable embedding layer, following the standard practice in ViT-based models [1].
>
>     [1] An Image is Worth 16x16 Words: Transformers for Image Recognition at Scale, ICLR 2021.
>
> - **Token Sequence Ordering**: To support autoregressive modeling, we arrange the patch tokens into a 1D sequence using a raster-scan ordering (left to right, top to bottom), consistent with the Vision Mamba design [2].
>
>     [2] Vision Mamba: Efficient Visual Representation Learning with Bidirectional State Space Model, ICML 2024.
>
> - **Mamba Encoder**: The backbone of RoMA is a Mamba-based encoder used for extracting image features, as stated in lines 216-217 of the manuscript.
>
>     Our contribution focuses on designing an effective **pretraining strategy tailored for remote sensing**, rather than altering the encoder’s architecture.
>
> - **Causal Cross-Attention**: During pretraining, we introduce a decoder to perform causal supervision via cross-attention. Specifically, the decoder comprises a sequence of virtual query tokens, each designed to predict a target patch token based only on encoder outputs from earlier positions (as defined by raster-scan order). This setup enables autoregressive learning while maintaining strict causality.
>
>     Although this cross-attention mechanism introduces quadratic complexity during pretraining (with respect to the decoder’s token length), we emphasize that:
>
>     > This decoder is only used during training to provide autoregressive supervision. It is discarded after pretraining. Thus, it does not affect the efficiency of the final foundation model at inference time.
>
> - **On Computional Complexity**: We respectfully believe there may be a misunderstanding. The foundation model we develop and ultimately deploy is the **Mamba encoder**, whose **linear complexity** [2] is one of its primary advantages:
>
>     $$
>     \mathcal{O}(N \cdot d)
>     $$
>
>     where $N$ is the number of tokens and $d$ is the hidden dimension. This contrasts with conventional ViT-based encoders used in many remote sensing foundation models, which exhibit quadratic complexity:
>
>     $$
>     \mathcal{O}(N^2 \cdot d)
>     $$
>
>     Hence, **our efficiency claims pertain strictly to the Mamba encoder**, and the decoder’s quadratic operations during training do not impact inference-time cost.
>
>     > **In summary**, RoMA's computational efficiency claims are based solely on the **Mamba encoder**, which is used at inference. The decoder contributes only during training and does not affect the model’s practical deployment cost.
>
>     [2] Vision Mamba: Efficient Visual Representation Learning with Bidirectional State Space Model, ICML 2024.
>
> - **Multi-Scale Prediction Strategy** Thank you for pointing this out. We clarify that the multi-scale prediction in RoMA is implemented by aggregating spatially adjacent decoder tokens and reconstructing their corresponding higher-level patches in parallel. As described in lines 241-243 of the paper.
>
>     Specifically, we construct **larger-scale features** by **grouping and concatenating adjacent decoded tokens** based on their spatial positions (e.g., 2×2 neighboring tokens for a 32×32 patch if the token represents 16×16), and then map these features back to pixel space using an MLP. This enables RoMA to predict target pixels at **multiple spatial granularities** (e.g., both 16×16 and 32×32 patch levels) simultaneously, thereby enhancing its ability to model objects with diverse scales in RS images.
>
>     We will revise both the text and Figure 3 in the updated manuscript to make this multi-scale strategy more intuitive and visually clear.
>
> ### **Q3:** The novelty of using a rotation augmentation with angular embedding for the pretraining method seems fairly limited
>
> **A3:**  Thank you for your comment. **Our contribution extends well beyond the use of rotation augmentation and angular embeddings.** Specifically, our goal is to address a fundamental challenge in current remote sensing foundation models: the inefficiency of ViT-based architectures in handling high-resolution inputs due to their quadratic complexity.
>
> To address this, we propose RoMA, a novel autoregressive pretraining framework built on a Mamba-based encoder with linear computational complexity, making it inherently more scalable to high-resolution remote sensing imagery.
>
> To support effective pretraining under the unique characteristics of remote sensing imagery, we introduce a series of tightly integrated strategies:
>
> - **(1) Sparsity of foreground objects:** We design an **adaptive region cropping strategy** to identify regions with high knowledge density, which often correspond to meaningful land objects. This ensures that the model focuses its learning capacity on informative regions.
>
> - **(2) Variation in object orientation:** We introduce **angular embeddings** that encode rotation angles into the token representations. This facilitates rotation-invariant feature learning and improves robustness to orientation diversity.
>
> - **(3) Extreme variation in object scales:** We propose a **multi-scale prediction strategy**, which reconstructs token representations at different spatial granularities (e.g., 16×16 and 32×32 patches), enabling the model to capture both fine-grained and coarse spatial patterns.
>
> These components are not isolated techniques, but are integrated cohesively into an autoregressive pretraining pipeline, which aligns well with Mamba’s sequential architecture. As a result, we successfully construct a remote sensing foundation model based on the Mamba framework, tailored to both the computational efficiency and unique characteristics of remote sensing imagery.

---

> > ### Comment · Reviewer_c7wu · 2025-08-03
> > **Comments on authors response**
> >
> > Thank you for you response. While the response helps in clarifying several aspects of the contribution, I remain critical of the overall clarity of the manuscript. I raised the scored accordingly.

---

> > > ### Author Response · Authors · 2025-08-03
> > >
> > > Thank you for your constructive feedback and for acknowledging our clarifications.
> > >
> > > We appreciate your emphasis on the overall clarity of the manuscript. We are committed to making improvements of the manuscript and thank you again  for your guidance in strengthening our work.

---

### Official Review · Reviewer_tojz · 2025-06-21

**Clarity:** 4
**Significance:** 4
**Originality:** 3
**Rating:** 6
**Confidence:** 5

**Summary:**

The authors propose rotation-aware multi-scale autoregressive learning (RoMA), a self-supervised pretraining method designed for Mamba-based architectures applied to remote sensing. The overall objective is to propose a first Mamba-based foundation model for remote sensing improving the scalability for high-resolution imagery compared to ViTs. To this aim, RoMa pretraining integrates a rotation aware pretaining objective and a multi-scale token prediction objective. Experiments have been performed for classification, change detection and semantic segmentation including ablation studies to evaluate technical choices on RoMA.

**Questions:**

Questions, comments and typos:

- Figure 1: please increase the size of all the text within the figure.

- Line 19, 71, 335: It is mentioned that extensive experiments have been conducted on "object detection". Shouldn't it be "change detection" instead (Line 305)?

- Related work: Please consider referring to CROMA, a foundation model mixing masked autoencoder and contrastive learning [5].

- Related work: Please consider referring to AnySat, a foundation model based on JEPA which could be considered as a MIM method [6].

- Figure 3, caption: What does "high-value patches" mean here? Not fined in the caption.

- Figure 4: "LBP" is not defined within the figure.

- Line 226: Missing hyperlink to Section 1.

- Line 239: "the autoregressive process compares them with their corresponding tokens in the original image for learning", it is not clear if the comparison correspond to the loss computation, which seems to be at the token level in this sentence, whereas clearly explained as at the pixel level in equation 2.

- Table 3 (b): Why selecting the LBP feature descriptor instead of HOG, where HOG shows better performance?
Please consider moving the figures either at the top or bottom of each page for better readability.

**Ethical Concerns:**

["NO or VERY MINOR ethics concerns only"]

**Final Justification:**

The authors' submission is high quality and will have a significant impact in the field of remote sensing. Additionally to all strengths mentioned in my review, the authors provided additional experiments on token vs. pixel space losses, comparisons with robust foundation models, and provided standard deviations demonstrating their method's performance and robustness. The Mamba-based architecture is expected to significantly impact the community by shaping research towards larger and/or higher resolution inputs.

**Limitations:**

yes

**Quality:**

3

**Strengths And Weaknesses:**

Strengths

- The overall submission is well written and well structured.

- Exploring a Mamba-based architecture as a remote sensing foundation model is relevant to tackle the ViT limitation of complexity bottleneck in attention layers w.r.t. large input size.

- The MAE pretraining strategy is commonly employed to pretrain remote sensing foundation model and the authors justify well in 3.2.1. its limitation, especially for a Mamba-based architecture, compared to an autoregressive setting.

- The adaptative rotation encoding strategy serves to mitigate a well known problem in remote sensing and includes learnable angle embeddings.

- A proper analysis (Figure 5) shows that a Mamba-based architecture along the proposed pretraining strategy scales well with the data volume and model size highlighting the potential of the work for future large-scale applications.
- Exhaustive experiments show better results compare to competing methods on well explained settings, including ablation studies to highlight the impact of each component and choices in RoMA.

Weaknesses

- The literature on rotation-invariant training for remote sensing, especially object detection, is rich [1, 2, 3] and could have been included to better understand if this strategy has already been used for pretraining or only for supervised learning.

- The multi-scale prediction strategy is similar to the multi-level loss adopted in Scale-MAE [4], one  would appreciate a clear comparison to state the novelty of the proposed strategy defined as a contribution.

- The loss function is computed in the pixel space whereas literature in MAE (see reference in related work) focuses the loss in the token space. One would appreciate to clearly state why the loss is not applied in the token space. Is this choice possible within the given framework? If yes, what would be the difference in the training process in terms of performances?

- Even if the selected methods are relevant for comparisons in Table 2, one may appreciate comparisons with more recent approaches that already shown the limitation of MAE-only based pretraining. In particular, one may include CROMA [5] pretrained with a combination of MAE and contrastive learning, or AnySat [6]  showcasing a JEPA framework for remote sensing foundation models. Both have shown impressive results and would be relevant competing methods.
- Experiments comparing performance with competing method would gain robustness if standard deviations would be included, e.g. by simply changing the initialisation of the linear prob.

I would like to emphasize that this submission is already of high quality and could have a significant impact on the community. I would consider increasing my rating if the weaknesses are properly considered.

References:

[1] J. Han et al., Redet: A rotation-equivariant detector for aerial object detection. In CVPR 2021.

[2] X. Yang and J. Yan. Arbitrary-oriented object detection with circular smooth label. In ECCV 2020.

[3] J. Han et al., Align deep features for oriented object detection. In IEEE Transactions on Geoscience and Remote Sensing 2022.

[4] C. J. Reed et al., Scale-MAE: A scale-aware masked autoencoder for multiscale geospatial representation learning. In ICCV 2023.

[5] A. Fuller et al., CROMA: Remote Sensing Representations with Contrastive Radar-Optical Masked Autoencoders. In NeurIPS 2023

[6] G. Astruc et al., AnySat: One Earth Observation Model for Many Resolutions, Scales, and Modalities. In CVPR 2025.

---

> ### Author Rebuttal · Authors · 2025-07-31
>
> ### Q1. Literature on rotation-invariant training could been included.
> We will incorporate relevant citations and expand the discussion accordingly in the revised version of the paper. Specifically, lines 175-182 will be revised as follows (new content highlighted in bold):
> *RS images often contain redundant airport runway pixels, while airplanes appear at various orientations, leading to different postures and shapes. **Such directional diversity has been extensively addressed in supervised object detection through rotation-equivariant or rotation-invariant designs~\cite{han2021redet, yang2020arbitrary, han2022align}.** However, autoregressive...*
>
> ---
> ### Q2. Difference between Scale-MAE with multi-scale prediction strategy
> While both RoMA and Scale-MAE involve multi-scale reconstruction strategies, their mechanisms and motivations are fundamentally different.
>
> **Scale-MAE** employs a Laplacian-pyramid-style decoder that progressively upsamples feature maps to produce multi-level representations. Each decoder stage is tasked with reconstructing the low-frequency or high-frequency component of the original image. The corresponding supervision consists of low-resolution and high-resolution versions of the input image, approximating its low- and high-frequency content, respectively.
>
> In contrast, **RoMA's multi-scale prediction strategy** does not rely on progressive decoding. Instead, it forms **larger-scale features** by **concatenating decoder token representations that are spatially adjacent** in the original image layout. This process enables RoMA to effectively reconstruct the input at **multiple patch granularities**, such as 16×16 and 32×32 patch-level representations, in parallel.
> In the revised version of the paper, we will provide a clearer comparison with Scale-MAE to highlight the novelty of RoMA's multi-scale design.
>
> ---
> ### Q3.Clearly state why the loss is not applied in the token space. Where possible within the given framework?
> We fully acknowledge that applying reconstruction loss in token space, as opposed to pixel space, has been shown in prior works such as BEiT[1] and CAE[2] to improve semantic representation learning in MAE-style frameworks. This is largely because token-space loss operates on semantically rich latent features, rather than low-level raw pixel intensities.
>
> However, such approaches typically depend on a pre-trained tokenizer (e.g., dVAE[3], CLIP[4]) or an auxiliary feature encoder to provide supervision in the latent space. In our work, we deliberately adopt pixel-space reconstruction loss in order to isolate the contribution of our RoMA framework itself, without introducing additional variables such as tokenizer quality or auxiliary networks. This design choice ensures a fair and transparent comparison with existing baselines under a unified pretraining setting.
>
> That said, token-space loss is fully compatible with our framework. To demonstrate this, we conducted a preliminary experiment comparing pixel-space and token-space reconstruction objectives. For the token-space setup, we adopt a frozen CLIP model as the teacher, and apply LayerNorm to normalize its output features. The RoMA framework serves as the student, augmented with a lightweight MIM head (fully connected layers) to predict the teacher’s latent representations. Both teacher and student receive the same full input image, and we compute the Smooth-L1 loss between their latent features across all spatial positions. The overall architecture is illustrated below:
> ```
> Input ──► Teacher (Frozen CLIP) ─► LN ─┐
>         │                              ├─► Smooth-L1 Loss (all tokens)
> Input ──► Student ───► MIM Head ───────┘
> ```
> This setup enables supervision in the token space, allowing the student to align with the high-level semantic representations produced by a strong teacher encoder. Compared to pixel-level objectives, this strategy encourages the model to learn more abstract and transferable visual features during pretraining.
> Due to time constraints, we pretrained both versions (pixel-space and token-space) for 50 epochs, and transferred the resulting models to the UCM and AID remote sensing classification benchmarks, following the same protocol as in our main paper. The results are as follows:
>
> |Pretraining Loss Type|UCM-55(%)|AID-55(%)|
> |-|-|-|
> |RoMA-Base(Pixel Space)|52.39|80.34|
> |RoMA-Base(Token Space)|56.67|81.72|
>
> These results confirm that token-space loss indeed yields stronger performance, further validating its effectiveness. More importantly, this experiment illustrates the flexibility and compatibility of the RoMA framework with different pretraining objectives.
>
> We consider this a promising future direction. To maintain a focused and methodologically clean evaluation of our core contributions, we plan to include this experimental comparison, along with implementation details, in the supplementary material of the revised version.
> [1] BEiT: BERT Pre-Training of Image Transformers, ICLR.
> [2] CAE: Context AutoEncoder for Self-Supervised Representation Learning, IJCV.
> [3] DVAE#: Discrete Variational Autoencoders with Relaxed Boltzmann Priors, NeurIPS.
> [4] Learning Transferable Visual Models From Natural Language Supervision, ICML.
>
> ---
> ### Q4. Comparisons with more recent approaches that already shown the limitation of MAE-only based pretraining, like CROMA and AnySat.
> We acknowledge the importance of including recent advances such as CROMA and AnySat, and we will incorporate citations, discussions, and comparisons with these works in the revised version of the paper. Specifically, lines 98–102 of the first paragraph in the related work section will be revised as follows:
>
> *MA3E integrates remote sensing angle factors into the MIM training architecture for both the encoder and decoder. **Recent efforts have also explored hybrid pretraining strategies that combine reconstruction and contrastive objectives. CROMA \cite{CROMA} jointly leverages masked autoencoding and cross-modal contrastive learning over spatially aligned SAR and optical imagery, aiming to improve both unimodal and multimodal representations. AnySat\cite{AnySat}, in contrast, focuses on learning a universal model across heterogeneous inputs by incorporating contrastive learning into a joint embedding predictive architecture, demonstrating strong performance on multi-resolution and multi-modal datasets**. While these methods...*
>
> To further address this point, we conducted preliminary comparisons. We have completed the evaluation of AnySat and CROMA on scene classification datasets using the same protocol as our main paper.  However, CROMA is tailored for multimodal SAR-optical inputs with a backbone designed for multispectral imagery. To adapt it, we added a 1×1 convolution to project the 3-channel RGB input into the 12-channel space required by CROMA, with the layer updated during linear probing.
>
> |Datasets|AID-55(%)|UCM-55(%)|
> |-|-|-|
> |CROMA|73.46|50.48|
> |AnySat|79.76|53.18|
> |RoMA-Base|87.36|59.45|
>
> These results highlight the strong performance of our RoMA framework. We also note that AnySat and CROMA may not perform optimally in this setting, possibly due to its pretraining being oriented toward multi-resolution, multi-sensor data, whereas AID and UCM are single-modal RGB datasets. With more extensive fine-tuning and adaptation, its performance may improve further.
>
> ---
> ### Q5. Experiments about standard deviations, e.g. linear prob
> Following your advice, we conducted linear probe classification experiments **three times** using different random seeds for initialization, while keeping all other settings consistent with the protocol described in our main paper. The results are summarized in the table below.
>
> |Methods|AID-55|UCM-55|
> |-|-|-|
> |ARM|81.07±0.24|50.14±1.49|
> |SatMAE++|86.14±0.51|55.93±2.13|
> |RoMA-Base|87.25±0.16|59.95±1.25|
>
> As shown, **RoMA exhibits consistently strong performance with low variance**, indicating better robustness compared to other competing methods.
> Due to time constraints, we report standard deviations for two representative baselines (ARM and SatMAE++). We will include standard deviation results for additional methods in the revised version.
>
> ---
> ### Q6. Typos and Minor Issues
> 1. The font size in Figure 1 will be enlarged in the revised manuscript.
> 2. The term "object detection" will be corrected to "change detection" in the revised manuscript.
> 3. We will cite CROMA in the related work section and briefly discuss its relevance in the revised manuscript.
> 4. We will include a citation to AnySat in the related work section and briefly discuss its connection to masked image modeling methods in the revised manuscript.
> 5. The term "high-value patches" refers to image patches with high information density, as computed using the LBP metric described in lines 180-187. In the revised version, we will provide a clearer explanation of this term in the caption of Figure 3.
> 6. In the revised version, we will include the full name of LBP (Local Binary Pattern) in the caption of Figure 4 for clarity.
> 7. The missing hyperlink to Section 1 will be added in the revised manuscript.
> 8. We acknowledge that the wording in line 239 may be misleading. To clarify, the comparison described indeed refers to the pixel-level loss computation, as explicitly defined in Equation 2. The phrase "corresponding tokens in the original image" simply refers to the decoder outputs being aligned with their corresponding image regions, rather than indicating token-level supervision. We will revise this sentence for clarity in the updated version.
> 9. We apologize for the mistake: the accuracy values for HOG and LBP were mistakenly swapped in Table 3(b). This will be corrected in the revised version of the manuscript. We also appreciate your suggestion regarding figure placement, and we will adjust the layout so that figures appear at the top or bottom of each page to improve readability.
>
> ---
> **Thank you again for your constructive feedback.**

---

> > ### Comment · Reviewer_tojz · 2025-08-01
> >
> > The Reviewer thanks the authors for their detailed rebuttal, which addressed all weaknesses, comments, and questions, and provided new insightful results, justifying an increased initial rating. The Reviewer suggests including additional experiments on pixel vs. token space optimization, comparisons with recent foundation models, and standard deviations in the final paper, as these results highlight the method's performance and robustness, and indicate interesting perspectives for future work.

---

> > > ### Author Response · Authors · 2025-08-02
> > >
> > > Thank you again for recognizing our work. In the final version, we will incorporate all three experiments. We sincerely appreciate your insightful feedback. Your support is a tremendous source of encouragement for our team.

---

### Official Review · Reviewer_9Qb3 · 2025-07-02

**Clarity:** 3
**Significance:** 4
**Originality:** 3
**Rating:** 5
**Confidence:** 5

**Summary:**

The paper introduces Mamba into the field of remote sensing imagery, utilizing autoregressive pretraining to construct a foundational model. It proposes a rotation-aware pretraining mechanism and multi-scale token prediction. The contributions of this study are as follows:
1)	This paper introduce RoMA, the first self-supervised autoregressive pretraining framework for Mamba architectures in remote sensing, enabling efficient scaling to high-resolution RS imagery.
2)	It propose a dynamic rotation-aware mechanism that integrates adaptive region cropping and angle-aware embeddings. By guiding the model to predict angular variations during autoregressive learning, it effectively addresses rotational diversity and sparse target distributions, enhancing rotation-invariant feature learning.
3)	It design a multi-scale prediction objective that addresses the extreme variations in object scales, enabling the model to learn robust object representations for downstream tasks.

**Questions:**

1. Remote sensing targets are typically small. Could the autoregressive pre-training method lead to the neglect of local information, thereby resulting in insufficient learning of small targets?
2. How can we ensure that the selection of top k tokens includes the target? If the target is not in a relatively central position, it may still lead to the situation shown in Figure 4(b).
3. In Figure 3, were larger blocks reconstructed using different orders?
4. Was angle prediction included? How does angle embedding function within the network?

**Ethical Concerns:**

["NO or VERY MINOR ethics concerns only"]

**Final Justification:**

Thanks to the authors for the detailed response.  I will keep my scores.

**Limitations:**

yes

**Quality:**

3

**Strengths And Weaknesses:**

Strengths : The idea is novel, and the theoretical reasoning is correct.
Weaknesses: Some parts of the text are unclear.

---

> ### Author Rebuttal · Authors · 2025-07-31
>
> ### **Q1. Remote sensing targets are typically small. Could the autoregressive pre-training method lead to the neglect of local information, thereby resulting in insufficient learning of small targets?**
>
> **A1:** We agree that small object representation is a critical challenge in remote sensing. However, we believe that the potential loss of local information is not inherently caused by the autoregressive pretraining paradigm, but is more related to the choice of patch size and whether the target region is adequately included in the input.
>
> To assess the ability of our RoMA model to capture small targets, we fine-tune RoMA-Base with UperNet [1] on the iSAID dataset [2], which contains a large number of small target categories. We compare its performance with several state-of-the-art models, reporting both the overall mIoU and the IoU for representative small-object categories, including Ship, Small Vehicle, Swimming Pool, and Plane. The average object widths for these categories are around 15-53 pixels.
>
> The results are shown in the table below. Our method achieves the highest overall mIoU and also shows notable improvements on small-object classes, especially Small Vehicle (average width 15 pixels), where RoMA surpasses all other backbones. These results suggest that the proposed **adaptive region cropping strategy** effectively increases the visibility of small foreground objects during pretraining, allowing the model to learn more discriminative representations.
>
> Fine-tuning performance on iSAID (following RingMo [3] settings):
>
> |Method|Backbone|mIoU|Ship ($33^2$ px)|Small Vehicle ($15^2$ px)|Swimming Pool ($34^2$ px)| Plane ($53^2$ px)|
> |---|---|---|---|---|---|---|
> |UperNet|IMP-ResNet-50|61.9|65.9|48.8|44.5|83.8|
> |UperNet|SeCo-ResNet-50|57.2|63.9|44.8|9.3|83.3|
> |UperNet|RSP-ResNet-50|61.6|64.2|47.5|43.8|82.8|
> |UperNet|ViT-B+RVSA|63.8|68.9|51.9|46.7|85.6|
> |UperNet|ViTAE-B+RVSA|63.5|69.6|51.9|47.5|85.4|
> |UperNet|RingMo|67.2|73.5|51.2|48.9|85.7|
> |**UperNet**|**RoMA-B (Ours)**|**67.4**|**73.8**|**53.7**|**51.8**|**86.0**|
>
> [1] Unified Perceptual Parsing for Scene Understanding, ECCV 2018.
>
> [2] iSAID: A Large-scale Dataset for Instance Segmentation in Aerial Images, CVPR Workshops, 2019.
>
> [3] RingMo: A Remote Sensing Foundation Model With Masked Image Modeling, IEEE TGRS 2022.
>
> ---
> ### **Q2. How can we ensure that the selection of top k tokens includes the target? If the target is not in a relatively central position, it may still lead to the situation shown in Figure 4(b).**
>
> #### **(1) How can we ensure that the selection of top k tokens includes the target?**
>
> **A2-(1):** This is an interesting question. To ensure that the selected tokens include the target, we incorporate a **three-stage safeguard mechanism** (see lines 183-195 of the main text):
>
> **(1) Coarse localization via information density:** We first compute an information density map using LBP, which highlights regions rich in texture and structure, often corresponding to land-cover targets. This enables us to localize potential object regions while filtering out most of the uninformative background.
>
> **(2) Candidate region selection**: To further ensure that land objects are included, we fix the size of candidate regions and generate all possible regions that fully contain the previously identified coarse localization points. Among these candidates, we select the region with the highest average information density, under the constraint that it must exceed the global average of the entire image. This improves the likelihood of capturing informative targets.
>
> **(3) Adaptive region shrinking**:  If **none** of the fixed-size candidate regions selected in step (2) satisfies the threshold, i.e., its average information density is **not higher than** the global average of the entire image, we reduce the region size and regenerate all candidate regions that still include the coarse localization point. We then repeat the selection process described in step (2). This adaptive refinement increases the likelihood of isolating smaller yet informative targets that may otherwise be diluted by surrounding background.
>
> Through this three-stage safeguard, our framework maintains strong coverage of targets, mitigating the potential risk of information loss.
>
> #### **(2) If the target is not in a relatively central position, it may still lead to the situation shown in Figure 4(b).**
>
> **A2-(2):** Thank you for your question. To evaluate the performance of our proposed **adaptive region cropping strategy**, we conducted a quantitative analysis using the full SpaceNet V1 building segmentation dataset.
>
> Specifically, we uniformly rescaled all images and their corresponding segmentation masks to a fixed resolution of 224×224, and divided each image into non-overlapping 16×16 patches. We then applied our adaptive region cropping strategy to each image to select a representative region.
>
> The results show that random cropping yields only 38.56%, whereas our method achieves a **significantly higher foreground object capture rate of 75.09%**, indicating that in over three-quarters of the samples, the selected region overlaps with at least one labeled object in the segmentation mask. This demonstrates the effectiveness of our adaptive cropping strategy.
>
> Foreground Object Capture Accuracy (SpaceNet V1):
>
> | Cropping Strategy  | Accuracy (%)  |
> | ----------  | ---- |
> | Random   | 38.56 |
> | Ours    | 75.09 |
>
> This substantial improvement demonstrates that our adaptive cropping strategy effectively increases the likelihood of capturing foreground objects, which is crucial for training autoregressive models to learn semantically meaningful representations. While a 75% overlap rate does not guarantee full object inclusion, it still provides a much denser learning signal compared to random cropping, and we found it sufficient to yield superior performance over existing methods in downstream tasks.
>
> We acknowledge that further improvement is possible. In future work, we plan to investigate more advanced region selection strategies, such as leveraging auxiliary weak annotations, to further enhance target coverage and representation learning.
>
> ---
> ### **Q3. In Figure 3, were larger blocks reconstructed using different orders?**
>
> **A3:** Thank you for your question. When reconstructing larger blocks, we did not employ different ordering strategies. Instead, we followed the **spatial layout of the original image patches**. For example, a 2×2 block would be arranged as:
>
> [1 , 2] &nbsp;&nbsp;&nbsp; &nbsp;[1 , 2]
>
> [5 , 6]     $\Rightarrow$     [5 , 6]
>
>
>
> This is consistent with the description in lines 241-242 of the main text:
>
> *"Building on MSE loss function, we concatenate token representations from each decoder block at a higher scale following a predefined **raster order**"*.
>
> We appreciate your comment and will revise Figure 3 to more clearly reflect this ordering scheme for better clarity.
>
> ---
> ### **Q4. Was angle prediction included? How does angle embedding function within the network?**
>
> **A4:** Thank you for your question. RoMA does not include an explicit angle prediction task. Instead, angle embeddings are incorporated to provide **directional prior information**, which helps guide the network toward learning **rotation-invariant representations** during pre-training. These embeddings are injected alongside patch tokens to condition the model with orientation cues, without requiring supervision from angle labels. The effectiveness of this strategy has been demonstrated in MA3E [1].
>
> [1] Masked Angle-Aware Autoencoder for Remote Sensing Images, ECCV 2024.
>
> ---
> **Thank you again for your constructive feedback. If you have any further questions, please don’t hesitate to let us know, we would be happy to continue the discussion.**

---

> > ### Comment · Reviewer_9Qb3 · 2025-08-03
> >
> > thanks for your response. All my concerns are well addressed.

---

> > > ### Author Response · Authors · 2025-08-04
> > >
> > > We are glad to have addressed your concerns and sincerely appreciate your insightful feedback. Your support is a tremendous source of encouragement for our team. Please let us know if there is any other question.

---

### Official Review · Reviewer_ezvU · 2025-07-03

**Clarity:** 4
**Significance:** 4
**Originality:** 4
**Rating:** 5
**Confidence:** 5

**Summary:**

This paper builds a Mamba-based pretrained foundation model that addresses the quadratic computational complexity bottleneck of Transformer-based models, enabling more efficient processing of high-resolution remote sensing imagery. It introduces a rotation-aware pretraining strategy that combines adaptive cropping with angular embeddings to better handle sparsely distributed objects with arbitrary orientations, and proposes a multi-scale token prediction objective to cope with large variations in object scale. Furthermore, extensive experiments demonstrate the scalability of the proposed foundation model, showing that its performance improves consistently with increased training data and model parameters, highlighting its potential as a strong foundation model for remote sensing tasks.

**Questions:**

1.While the proposed framework shows promising results on standard optical RS datasets, could the authors comment on or provide evidence regarding the applicability of RoMA to other types of remote sensing data such as SAR or hyperspectral imagery? Since these modalities have very different data characteristics, it is important to clarify whether the method generalizes beyond the current experimental scope.


2.The paper claims advantages in handling high-resolution images. However, the highest resolution used in the experiments appears to be 1248×1248, which is relatively modest compared to typical operational remote sensing images that can reach tens of thousands of pixels per side. Could the authors either include experiments on higher-resolution datasets or clarify the limitations of the current approach in truly large-resolution scenarios?

**Ethical Concerns:**

["NO or VERY MINOR ethics concerns only"]

**Limitations:**

Despite its contributions, this work has a few limitations. First, the generalizability of the RoMA framework to other visual domains or non-optical remote sensing modalities, such as SAR or hyperspectral imagery, remains unverified. These modalities exhibit different spatial and spectral properties that may challenge the current design. Second, although the framework is designed to handle high-resolution images, the experiments are conducted on images up to 1248×1248 resolution, which may not reflect the challenges posed by ultra-high-resolution remote sensing data commonly used in real-world applications. Future work could explore extending RoMA to broader data modalities and scaling to much higher image resolutions.

**Quality:**

4

**Strengths And Weaknesses:**

Strengths:

1.It pioneers the use of the Mamba architecture in self-supervised learning for remote sensing, addressing the inefficiency of traditional Transformer-based models when handling high-resolution imagery.

2.By leveraging the linear computational complexity of Mamba, the proposed framework achieves significant improvements in scalability and computational efficiency, which are critical for large-scale remote sensing applications.

3.The paper introduces well-motivated and technically sound components—including a rotation-aware pretraining strategy and multi-scale token prediction—that are carefully designed to address the unique challenges of remote sensing data, such as orientation variability and scale diversity.

4.Extensive experiments across multiple downstream tasks (scene classification, object detection, and semantic segmentation) demonstrate consistent performance gains over ViT-based models, supporting the generality and robustness of the proposed method.

5. The work is aligned with the development of scalable and general-purpose RS foundation models, and the authors provide scaling law analyses that further validate its long-term potential.


Weaknesses:

1. It remains unclear how well the proposed RoMA framework generalizes to different types of remote sensing imagery such as SAR or hyperspectral data.

2.The authors claim that the proposed model has significant advantages in handling high-resolution remote sensing images; however, to my knowledge, remote sensing imagery typically has resolutions much higher than 1248×1248. The paper should consider evaluating on higher-resolution data to more convincingly demonstrate its superiority in high-resolution scenarios.

---

> ### Author Rebuttal · Authors · 2025-07-31
>
> ### **Q1. While the proposed framework shows promising results on standard optical RS datasets, could the authors comment on or provide evidence regarding the applicability of RoMA to other types of remote sensing data such as SAR or hyperspectral imagery? Since these modalities have very different data characteristics, it is important to clarify whether the method generalizes beyond the current experimental scope.**
>
> **A1:** Thank you for your insightful question. In principle, RoMA can potentially be extended to other types of remote sensing data such as SAR and hyperspectral imagery.
>
> For **SAR data**, although it differs from optical imagery in imaging mechanism (active microwave vs. passive optical), it still follows a similar top-down observation geometry. This allows the **rotation-invariant operations** and **multi-scale prediction strategies** in RoMA to remain applicable. Moreover, SAR images often exhibit high-intensity backscatter in regions corresponding to man-made structures (e.g., buildings, ships), which aligns well with RoMA's **adaptive cropping strategy** based on information density. In terms of channel compatibility, most SAR images contain 1–4 channels; for single-polarization data (e.g., VV), we can replicate the image into a 3-channel format to match the input requirements of the Mamba backbone.
>
> For **hyperspectral data**, although it consists of a large number of spectral bands, the core components of RoMA, can be extended to 3D spatial-spectral cubes. Inspired by SpectralGPT [1], the hyperspectral image cube can be partitioned into spatial-spectral subcubes. We can then compute a 3D information density map by evaluating each subcube's spatial saliency and spectral variance. Based on this map, the most informative subcubes can be selected, and **adaptive cropping strategy** can be applied to their surrounding 3D regions. The **multi-scale prediction strategy** can also be extended by considering multiple scales in both spatial and spectral dimensions. For the **rotation operation**, unlike traditional 3D image volumes, hyperspectral cubes consist of two spatial dimensions and one spectral dimension without a well-defined geometric structure across all three axes. Therefore, we do not apply geometric 3D rotations to subcubes. Instead, we enhance data diversity using hyperspectral-specific augmentations such as spectral band shuffling. These operations preserve the semantic consistency of spatial-spectral mappings while improving the model's generalization ability.
>
> In summary, while current experiments focus on standard optical datasets, the architecture and principles of RoMA are general enough to be extended to SAR and hyperspectral imagery.
>
> [1] SpectralGPT: Spectral Remote Sensing Foundation Model, IEEE TPAMI 2023.
>
> ### **Q2. The paper claims advantages in handling high-resolution images. However, the highest resolution used in the experiments appears to be 1248×1248, which is relatively modest compared to typical operational remote sensing images that can reach tens of thousands of pixels per side. Could the authors either include experiments on higher-resolution datasets or clarify the limitations of the current approach in truly large-resolution scenarios?**
>
> **A2:** Thank you for the valuable question. To address the reviewer's concern, we conducted additional experiments on higher-resolution images using RoMA-Base and ViT-Base. Each model was evaluated over 200 forward passes to ensure statistical reliability. We report peak GPU memory usage (in MB) and average inference speed (samples per second) across a range of input resolutions. All experiments were conducted on a single NVIDIA A100 GPU with a batch size of 1. (Note: Figure 1 in the main text uses an RTX 4090 GPU with a batch size of 2.)
>
> Peak memory usage (MB):
>
> | Resolution  | 768  | 1024  | 1248  | 1520  | 2048  | 3072  | 4096  |
> | ----------  | ---- | ----- | ----- | ----- | ----- | ----- | ----- |
> | RoMA-Base   | 2693 | 4504  | 6526  | 9434  | 16934 | 37485 | 66357 |
> | ViT-Base    | 4229 | 11726 | 24531 | 52106 | OOM   | OOM   | OOM   |
>
> Inference speed (samples/sec):
>
> | Resolution | 768   | 1024  | 1248  | 1520 | 2048 | 3072 | 4096 |
> | ---------- | ----- | ----- | ----- | ---- | ---- | ---- | ---- |
> | RoMA-Base  | 24.98 | 15.91 | 11.43 | 7.86 | 4.37 | 2.00 | 1.15 |
> | ViT-Base   | 22.11 | 9.94  | 4.99  | 2.57 | OOM  | OOM  | OOM  |
>
> These results demonstrate the scalability of RoMA to higher-resolution inputs. In contrast, ViT-Base encounters out-of-memory (OOM) errors at resolutions beyond 2048×2048. This highlights RoMA's superior computational efficiency and its potential applicability to operational remote sensing scenarios involving ultra-high-resolution images.
>
> ---
> **Overall, thank you again for your constructive feedback. If you have any further questions, please don’t hesitate to let us know, we would be happy to continue the discussion.**

---

> > ### Comment · Reviewer_ezvU · 2025-08-08
> >
> > The author's response somewhat addresses my concerns, but I still recommend adding experimental results to support their point. The article is innovative and demonstrates the scalability of Mamba as a foundational model, so I maintain my original score.

---

> > > ### Author Response · Authors · 2025-08-08
> > >
> > > Thank you again for recognizing our work.
> > > *  In the final version, we will include all high-resolution experiments to better demonstrate RoMA’s advantages at higher resolutions.
> > > *  Additionally, extending RoMA to other types of remote sensing data is our future work. We will pursue this direction and develop a multi-source version of RoMA.
> > >
> > > Your feedback is highly constructive and greatly enhances the quality of our paper. We sincerely appreciate your positive score and continued support for our work.

---

> ### Comment · Area_Chair_sMVC · 2025-08-04
> **post-rebuttal comments**
>
> Dear reviewer ezvU,
>
> Can you please share your thoughts post rebuttal, once you've had a chance to see the author responses and also the other reviews?
>
> Best,
> AC

---

### Decision · Program_Chairs · 2025-09-17

**Decision:**

Accept (poster)

**Comment:**

This paper presents a framework, RoMA, with the aim of working around the quadratic complexity of standard transformers such as ViT, in the context of remote sensing data, which includes high-resolution images. To this end, RoMA follows the Mamba architecture and shows how this can be adapted to a self-supervised learning scenario. Two key ingredients in achieving this are: (1) rotation-aware pretraining, and (2) multi-scale token prediction objectives.

Several strengths of this work are:
- bringing significant improvements in scalability and computational efficiency,
- showing the importance of this work in the context of remote sensing,
- exhaustive and convincing experimental validation and analyses.

After the discussion phase, the main weakness of the paper was identified as the lack of presentation clarity by reviewer c7wu, who gave a slightly lower rating of borderline reject. All the other reviewers were convinced by the paper, the rebuttal and the additional information provided in the discussion phase. Reviewer c7wu also acknowledged their low confidence in this topic (compared to the other reviewers) and was prepared to give the authors the benefit of doubt regarding the improvement of presentation clarity.

Thus, the area chair recommends the paper for acceptance. All the additional information and clarification presented in the rebuttal and the discussions should be included in the final version of the paper, with particular attention to presentation clarity.